# Beyond Denouncing Hate:
## Strategies for Countering Implied Biases and Stereotypes in Language

*Warning: content in this paper may be upsetting or offensive.*

**Jimin Mun**[♡]  **Emily Allaway**[♠]  **Akhila Yerukola**[♡]
**Laura Vianna**[†]  **Sarah-Jane Leslie**[◇]  **Maarten Sap**[♡][♣]

[♡]Carnegie Mellon University  [♠]Columbia University  [†]University of Washington

[◇]Princeton University  [♣]Allen Institute for AI

## Abstract

Counterspeech, i.e., responses to counteract potential harms of hateful speech, has become an increasingly popular solution to address online hate speech without censorship. However, properly countering hateful language requires countering and dispelling the underlying inaccurate stereotypes implied by such language. In this work, we draw from psychology and philosophy literature to craft six psychologically inspired strategies to challenge the underlying stereotypical implications of hateful language. We first examine the convincingness of each of these strategies through a user study, and then compare their usages in both human- and machine-generated counterspeech datasets. Our results show that human-written counterspeech uses countering strategies that are more specific to the implied stereotype (e.g., counter examples to the stereotype, external factors about the stereotype's origins), whereas machine-generated counterspeech uses less specific strategies (e.g., generally denouncing the hatefulness of speech). Furthermore, machine-generated counterspeech often employs strategies that humans deem less convincing compared to human-produced counterspeech. Our findings point to the importance of accounting for the underlying stereotypical implications of speech when generating counterspeech and for better machine reasoning about anti-stereotypical examples.

## 1 Introduction

Counterspeech, i.e., responses that counteract hateful or dangerous content (Benesch et al., 2016), has emerged as a widely supported solution to address hateful online speech without censorship risks of deletion-based content moderation (Myers West, 2018; Sap et al., 2019). However, as the scale of hateful content increases online (Leetaru, 2019; Baggs, 2021), effectively responding with counterspeech is infeasible for humans to do alone. As

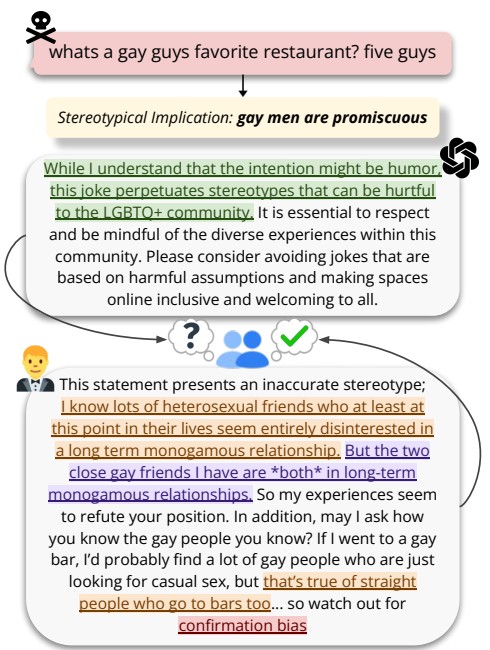

Figure 1: Responses to same implication where humans are able use appropriate countering strategies to target the implication by presenting *alternate groups*, *counterexamples*, and *external factors* while LLMs focus on *denouncing*.

such, NLP systems can provide a promising opportunity to further understand counterspeech and assist content moderators and other stake holders in generating responses (Parker and Ruths, 2023).

One major challenge towards effectively combating hateful and prejudiced content is that counterspeech should dispel the implied stereotypical beliefs about social groups (Buerger, 2021b), beyond simply denouncing the speech. For instance, in Figure 1, the human-written counterspeech targets the stereotype that "*gay men are promiscuous*" implied in the original post. If left unchallenged, such prejudiced implications can perpetuate and maintain existing stereotypes (Beukeboom and Burgers, 2019) and exacerbate discrimination and intergroup conflict (Fiske, 1998; Macrae and Bodenhausen, 2000). However, exactly how to convincingly combat stereotypical implications with counterspeech

remains an open question (Buerger, 2022).

To investigate this question, we analyze the usage and convincingness of psychologically inspired strategies for countering implied biases and stereotypes, in both human- and machine-generated counterspeech. In contrast to prior work, which has examined tone- or style-based effectiveness (Hangartner et al., 2021; Han et al., 2018), we design and analyze six *stereotype-targeting strategies* drawing from literature in social psychology, social cognition, and philosophy of language.

Our strategies aim to directly combat the stereotypical association between a *group* and an *attribute* (e.g., "gay men", "promiscuous"): reducing the association between the attribute and the group by suggesting *alternate groups* that exhibit similar attributes, or focusing on *different attributes* of the targeted group, providing *counterexamples* to the stereotype, reducing the intrinsic implication of the stereotype through *external factors* about stereotype origins, broadening the group boundaries to emphasize individual variation, and generally labeling the stereotype as negative.

We first investigate how prevalent and convincing these strategies are in human-generated counterspeech. Specifically, we conduct a crowdsourced user study to examine which strategies workers prefer, and analyze two corpora of human-written counterspeech. Our results show that humans perceive countering strategies that present different qualities of the targeted group as most persuasive, and generally prefer strategies that require deeper reasoning about the implied stereotype (e.g., presenting external factors or counterexamples).

Furthermore, when comparing human-written and machine-generated counterspeech, we find that machines often generate nonspecific strategies (e.g., general denouncing) whereas humans use more nuanced and specific strategies, like counterexamples. Our findings highlights the potential for understanding counterspeech strategies that persuasively challenge stereotypical beliefs and reveal the inherent limitations of current NLP models in generating effective stereotype-targeting counterspeech.

## 2   Background: Language, Stereotypes, and Counterspeech

Our goal is to investigate stereotype-targeting counterspeech strategies that most convincingly challenge the underlying implications of hate speech. Our analyses and strategies are based on the fact that prejudiced language about social groups can often convey and exacerbate stereotypes through *subtle implicature* (Fiske, 1998; Sap et al., 2020; Perez Gomez, 2021, §2.1). Properly combating such language requires convincingly dispelling and fact-checking the stereotypical implications beyond simple deflection or norm-setting (Lepoutre, 2019, §2.2). Additionally, our investigations aim to shed light on how NLP systems can help humans better analyze and produce counterspeech (§2.3).

### 2.1   Language and Stereotypes

Prejudiced language reflects and perpetuates social inequalities and stereotypical beliefs (Beukeboom and Burgers, 2019). As argued by Perez Gomez (2021), when accepting subtly prejudiced statements as true, we inherently accept the underlying stereotype as true. Therefore, the harmful implications of hate speech must be challenged to mitigate the spread of such beliefs.

These underlying stereotypes are often in the form of *generics*, i.e., generalizing statements such as "girls wear pink", (Rhodes et al., 2012) which are difficult to counter due to their unquantified structure (Leslie, 2014). Generics show especially strong association to the group when the characteristics are striking and negative (Leslie, 2017), and once formed, these beliefs become extremely difficult to change (Scheffer et al., 2022). Thus, countering hate speech is important yet challenging because it requires extensive reasoning about the underlying implications.

### 2.2   Countering Hate vs. Implied Stereotypes

Counterspeech can have many intended effects to a diverse audience including original speakers and bystanders (Buerger, 2021a, 2022). Therefore, counterspeech, and more broadly countering hate, has been studied from a number of different angles. Observing Twitter interactions, Hangartner et al. (2021) found that empathy resulted in better outcomes (e.g., deletion, decrease in number of hate tweets) compared to using humor or warnings of consequences. Benesch et al. (2016) suggests various strategies for countering hate such as empathy, shaming, humor, warning of consequences, and fact-checking. Moreover, studies show the contagion effect (Buerger, 2021a) where social feedback (Berry and Taylor, 2017) and the quality of the comments (e.g., civility) impact subsequent comments and discussions (Friess et al., 2021; Han et al., 2018). These works provide insights into

preferences for strategies to counter the explicit content of hate speech. In contrast, our work studies strategies for persuasive arguments against hate conveyed through *implied* stereotoypes.

To specifically target stereotypes and group perception, previous works in psychology have studied humanization of targeted groups through reminding people that individuals belong in multiple categories (Prati et al., 2016), exposure to positive outgroup examples (Cernat, 2011), and anti-stereotypes (Finnegan et al., 2015) and have shown promising results. Moreover, the positive impacts on belief change of fact-checking (Porter and Wood, 2021) and counter narratives (Lewandowsky et al., 2012) call for counterspeech beyond just rejecting the original speech.

## 2.3 NLP for Counterspeech

With the ability to analyze and generate text at a large scale, NLP is an invaluable asset for counterspeech and number of works in NLP have collected and generated counterspeech datasets (Mathew et al., 2019; Garland et al., 2020; Qian et al., 2019; Chung et al., 2019; Bonaldi et al., 2022; Fanton et al., 2021) for the detection, analysis, and generation of counterspeech. Yu et al. (2022) observed that question marks and words related to awareness and problem solving are frequently used in counterspeech. Similarly, Lasser et al. (2023) noted that responding with simple opinions or sarcasm reduced extreme hate. Generative approaches to counterspeech have incorporate stylistic strategies (e.g., politeness, joyfulness, detoxification; Saha et al. 2022) and relevance (Zhu and Bhat, 2021). Although previous works have made progress towards observing and generating using different counterspeech strategies, the focus has been on stylistic differences and while promising, do not explcitly target the underlying stereotypes in hate speech.

Fraser et al. (2021) and Allaway et al. (2023) have more closely investigated computational methods to counter stereotypes. Fraser et al. (2021) investigated stereotypes and counter stereotypes using warmth and competence of the groups and associated words and noted the difficulty in automatically generating anti-stereotypes. Moreover, Allaway et al. (2023) have studied automatic generation of counterstatements to implied stereotypes and provides a proof-of-concept for countering strategies. Our work, on the other hand, adds upon the proposed strategies and analyzes their usage by

| |
|---|
| **Post**: Do girls think guys flex muscles n stroke their c\*\*ks before taking pics Bc that's basically what pathetic girls do in every pic |
| **Implied Stereotype**: *Women are shallow.* |
| **ALT-GRP** This statement implies an inaccurate stereotype because many men can also be shallow. |
| **ALT-QUAL** This statement implies an inaccurate stereotype because many women are intelligent, profound, and multi-dimensional. |
| **CNTR-EXS** This statement implies an inaccurate stereotype because a lot of women are profound, including women philosophers, women scientists, and women writers. |
| **EXTERNAL** This statement implies an inaccurate stereotype; historical contexts such as gender inequality and sexism have perpetuated this false stereotype that women are shallow. |
| **BROAD** This statement implies an inaccurate stereotype because women have all sorts of personality types and characteristics, just like all groups of people; we cannot characterize them all as shallow. |
| **GEN-DEN** This statement's implication is an inaccurate and unnecessarily hurtful generalization about women that they are shallow. |

Table 1: Example statements of each countering strategy. These statements were also used as counter statements for user study in §4.

humans and machines to understand these strategies and improve counterspeech. Building on these prior works, we aim to provide insights into how language models can help counter stereotypes and mitigate the harmful effects of hate speech.

## 3 Stereotype-Targeting Strategies

To investigate how counterspeech can address stereotypical speech, we examine stereotypes through the lens of *generics* about social groups (Rhodes et al., 2012; Sap et al., 2020), i.e., short unquantified statements that link a social group with a quality:

$$\text{GROUP} + relation + \text{QUALITY}.$$

For example, the stereotype in Table 1 "Women are shallow" can be separated into three components: "Women" (GROUP), "are" (*relation*) and "shallow" (QUALITY). The GROUP typically refers to demographic groups such as black people, gay men, etc. The *relation* is typically a stative verb like "are" or a verb indicating a lack of ability, such as "can't". A QUALITY usually has negative meaning or connotation (e.g., shallow, terrorist), but can sometimes be positive (e.g., for positive stereotypes; Cheryan and Bodenhausen, 2000).

Through this formulation, we introduce six countering strategies that aim to *challenge the acceptability, validity, or truthfulness of stereotypes* ex-

pressed as generics, crafted based on prior work on combating stereotypes and essentialism in social psychology and finding exceptions to generics in philosophy of language.

**Alternate Groups (ALT-GRP)**   People often *accept* generics of the form '*G relation q*' as true based on relatively weak evidence (Cimpian et al., 2010). However, for unfamiliar groups, we often attribute qualities more assertively, claiming that (almost) '*all G relation q*' even with scarce supporting evidence (Rooij and Schulz, 2020). To counter such inferences, we can remind readers that alternate groups' relation to $q$ is often higher than commonly recognized.

We define the ALT-GRP strategy as follows:

$$Alt(\text{GROUP}) + relation + \text{QUALITY}$$

Here, $Alt(G)$ represents a relevant set of alternative groups to $G$ with the same set of separating characteristic (e.g., gender). For instance, if $G =$"women", $Alt(G)$ would consist of a group like "men" rather than a set of unrelated groups.

**Alternate Qualities (ALT-QUAL)**   Similarly, we construct a strategy that highlights alternate distinctive or defining qualities of $G$. We consider values of $Alt(q)$ to promote a more nuanced understanding and challenge preconceived notions about the group. Thus, we formulate the ALT-QUAL strategy as the following:

$$\text{GROUP} + relation + Alt(\text{QUALITY})$$

In this case, $Alt(q)$ represents alternate contextually relevant qualities. For example, if $S =$ Women are shallow., $Alt(q)$ would include qualities like "*intelligent*" and "*profound*" but not positive and unrelated quality (e.g., fun). This aims to combat the definitional aspect of a stereotype.

**Counterexamples (CNTR-EXS)**   Counterexamples aim to point out exceptions to the stereotype and thereby challenge the implication that the stereotype holds for all members of the GROUP.[1]

$$x + \begin{cases} relation + Alt(\text{QUALITY}) \\ \neg relation + \text{QUALITY} \end{cases}$$

where $x$ is a specific individual member of GROUP or a subgroup of GROUP[2].

---

[1]This strategy follows a similar formulation as the direct exception statements in Allaway et al. (2023).

[2]A subgroup must be precisely defined; for example, "my gay friends" is specific, while "most Muslims" or "gay people 30 years ago" are not.

**External Factors (EXTERNAL)**   When readers are reminded that stereotypical characteristics of a group $G$ exist due to external conditions (i.e '*G relation q is due to external factor f*'), they have been shown to exhibit more flexible and dispensable thinking towards the group $G$ (Vasilyeva and Lombrozo, 2020). This style of unconfounding is referred to as *structual construal* (Berkowitz, 1984) in social psychology.

Thus, we propose the EXTERNAL strategy, in which counterspeech employs external causes or factors to demonstrate why a stereotype might have been formed (e.g., historical context such as gender inequality). Note that it differs from CNTR-EXS by focusing on external factors or details for refutation, rather than specific instances within the group.

**Broadening (BROAD)**   The broadening strategy questions the validity of a GROUP in a generic or stereotype by employing humanizing techniques. Essentially, broadening emphasizes either the uniqueness of a certain quality to the group or highlights the complexity of individual group members, suggesting that they cannot be defined by merely one quality.

An example of broadening can be formalized as:

$$\text{All people} + relation + \begin{cases} Alt(\text{QUALITY}) \\ \text{QUALITY} \end{cases}$$

**General Denouncing (GEN-DEN)**   Another commonly observed countering strategy in NLP is denouncing (Mathew et al., 2019; Qian et al., 2019). Denouncing involves expressing general disapproval for the hate speech (e.g., "This is just wrong"). This approach has the potential to establish discursive norms and preemptively dismiss similar instances of hate speech (Lepoutre, 2019). Here, counterspeech assigns general negative values to the stereotypical statement, using terms like "hurtful", "unhelpful" or "not okay" to express disapproval and challenge the stereotype.

## 4   Human Evaluation of Stereotype-Targeting Strategies

In order to determine which strategies are persuasive in countering implied stereotypical beliefs, we first conduct a user study on Amazon Mechanical Turk (MTurk) to measure the convincingness of expert-crafted *counter-statements*, i.e., short sentences that embody a specific countering strategy.

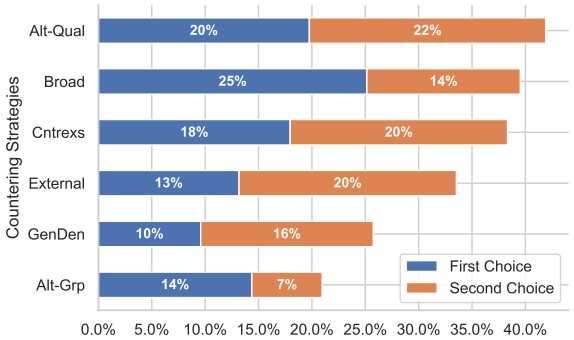

Figure 2: First and second choices of countering strategies sorted by the sum of the two responses. Percentage is calculated separately for first and second choice.

## 4.1 Experimental Setup

We first collected social media posts with their corresponding implied stereotype for 10 demographic groups from the Social Bias Inference Corpus (SBIC; Sap et al., 2020). The groups were selected from the top 25 most targeted groups in SBIC[3], and the stereotypes were selected from the top 10 most frequent stereotypes for each group. The selected groups and stereotypes are shown in Figure 7.

**Counter-statements** For each stereotype, six counter-statements (one for each strategy) were crafted collaboratively by the authors, to ensure maximal faithfulness to the strategy definitions. This resulted in a total of 60 counter-statements; see Table 1 for examples.

**Task Description** Each participant is presented with a post and its implied stereotype, along with six counter-statements in random order. Annotators are asked to select the first and second most convincing counter-statements according to them (i.e., following a descriptive data labeling paradigm; Röttger et al., 2022). We recruited annotators from a pool of pre-qualified participants who were paid 0.27 USD for each text and stereotype pair. Please refer to Appendix G for further details.

## 4.2 Results

Our results in Figure 2 suggest that there is a clear distinction and ordering of convincing countering strategies. Although the order of preference for the first choice and the second choices are different, the cumulative top choices remain consistent:

---

[3]The groups were aggregated by a rough regex matching identity terms, for example, "asian [folks|people|person]" or "asians" for Asian folks, within the target group identified by the SBIC annotators.

ALT-QUAL, BROAD, CNTR-EXS, and EXTERNAL. These results are mostly consistent with findings from social psychology that highlight the value of strategies similar to ALT-QUAL (Leslie, 2008, 2017), BROAD (Foster-Hanson et al., 2022), and EXTERNAL (Vasilyeva and Lombrozo, 2020).

We also observe that less complex strategies are not necessarily the most convincing In particular, while GEN-DEN and BROAD require the least amount of reasoning about the stereotypical quality or group to understand, they have very different levels of convincingness: BROAD is the *most* preferred first choice while GEN-DEN is the *least* preferred. Furthermore, strategies that require a similar level of involved reasoning can have drastically different levels of convincingness (e.g., ALT-QUAL is chosen as convincing twice as frequently as ALT-GRP, cumulatively). This suggests that differences in strategy complexity are not sufficient to account for convincingness.

There are also clear differences in strategy preferences depending on the group and stereotype (see Figure 7). For example, BROAD is not chosen at all for the stereotype about feminists (Fig. 7, fourth from the bottom), despite its popularity in countering other stereotypes such as "women are shallow".

## 5 Countering Strategies in Human vs. Machine-generated Data

The results from the previous section suggest that some strategies are more convincing than others. To further investigate this, we examine the usage of these six strategies in human-written counterspeech. Additionally, we compare human and machine-generated counterspeech to understand whether machines use these strategies. In this section, we discuss the methods and datasets used for this analysis, while our findings are presented in §6.

## 5.1 Datasets

We consider two datasets that contain *posts* paired with counterspeech *comments*: Multitarget-CONAN (Fanton et al., 2021), a dataset of hate speech and counterspeech pairs generated using human-machine collaboration, and the Winning Arguments (ChangeMyView) Corpus (Tan et al., 2016) extended with more recent posts from the same subreddit, r/ChangeMyView (hence forth CMV) for the period 2016-2022.[4] Appendix A

---

[4]https://www.reddit.com/r/changemyview/

shows details on datasets and their collection and filtering process.

These two datasets offer insights into how countering strategies are employed in two distinct types of online interactions: in CONAN, users share *short comment* replies without necessarily seeking change, while in CMV, users actively engage via *long comment* posts to challenge or potentially alter their opinions.

## 5.2 Selecting Data

For both MTCONAN and CMV datasets, we are interested in analyzing the implied stereotype of the original post as well as the strategies used in the counter-statements (counterspeech in MTCONAN; $\Delta$ and non-$\Delta$ comments in CMV). We use the process outlined below to filter and select data that contain specific implied stereotypes.

**Implied Stereotype Extraction** To analyze strategies for combating stereotypical implications, we filter our two datasets to include only posts that target or reference a specific stereotype about a demographic group. Based on evidence of its labeling abilities (Ziems et al., 2023), we use GPT-4 to identify such posts and generate the target group and implied stereotype, by prompting it with three in-context examples (see Appendix B for details). Table 3 shows the number of unique harmful posts in each dataset.

## 5.3 Counterspeech Generation

To investigate machine-generated counterspeech, we prompt two large language models (LLMs) to respond to posts with counterspeech: GPT-4 (OpenAI, 2023) and Alpaca (Taori et al., 2023). We instruct the LLMs to assist a user in responding to harmful posts without providing any specific counter strategy definitions, to avoid biasing the model's output towards predetermined approaches.[5]

## 5.4 Counter Strategy Labeling

To detect the usage of any our six stereotype-targeting strategies within a reply to a post, we leverage the labeling abilities of two LLMs, namely GPT-3.5[6] and GPT-4. We prompt these models

---

[5] We provide the LLM with the prompt, "*How should I respond to a post that says [POST]*," along with the system message, "*You are helping people respond to harmful posts online. Reply directly to the post.*" See Appendix D for further details.

[6] gpt-3.5-turbo-0314

with the strategy definition and two in-context examples (one using the strategy and one without it), and instruct them to provide a binary label that indicates the presence of the strategy. We also prompt them to output the text span that contains it. Please refer to Appendix D.2 for detailed prompts used for analysis.

**Test Set Creation** To validate the reliability of using LLMs for detecting these nuanced countering strategies, we randomly sampled 100 examples from each dataset. These samples were assigned gold labels through expert annotation conducted by three of the authors. The authors manually reviewed the examples independently, resolving disagreements through discussions. This process led to the creation of a test set comprising of 200 expert-annotated samples.

**Evaluating Strategy Detection** We utilize these gold labels to assess the performance of each LLM in detecting strategies, as shown in Table 2. Although performance is only moderate, both models are outperform the majority class classifiers, which include both the 0 and 1 majority classes. This finding is particularly notable given that the majority class of the entire dataset is unknown. It suggests that these models have recognized some meaningful signals within the data. GPT-4 shows similar or improved performance on strategy detection compared to GPT-3.5 for all strategies and is thus used in subsequent experiments for strategy labeling. However, akin to expert-annotators, determining whether a reply contains a specific strategy is not trivial for LLMs. For example, we observed during a qualitative inspection that models occasionally confuse simple mention of alternative qualities or groups for the presence of ALT-QUAL or ALT-GRP, respectively. See Appendix F for further analysis.

## 6 Usage and Convincingness Results

In this section, we discuss the results of our investigations regarding the usage and persuasive power of our six stereotype-targeting strategies, for both human and machine-written counterspeech.

## 6.1 CMV: Persuasive vs. Non-persuasive Comments

The comparison of strategy usage between $\Delta$ and non-$\Delta$ comments in CMV dataset is shown in Figure 3. Overall, the most frequently used strategies were ALT-QUAL (84.8%) followed by EXTERNAL

| Method Strategy | GPT-4 Precision | Recall | F1 | GPT-3.5 Precision | Recall | F1 | Majority Class (0) Precision | Recall | F1 | Majority Class (1) Precision | Recall | F1 |
|---|---|---|---|---|---|---|---|---|---|---|---|---|
| ALT-GRP | **0.64** | **0.75** | **0.60** | 0.62 | 0.71 | 0.59 | 0.42 | 0.5 | 0.45 | 0.09 | 0.5 | 0.15 |
| ALT-QUAL | **0.68** | **0.66** | **0.61** | 0.66 | 0.64 | 0.58 | 0.31 | 0.5 | 0.38 | 0.20 | 0.5 | 0.28 |
| CNTR-EXS | **0.74** | **0.80** | **0.73** | 0.73 | 0.78 | 0.72 | 0.36 | 0.5 | 0.42 | 0.14 | 0.5 | 0.22 |
| EXTERNAL | 0.69 | **0.72** | **0.68** | **0.70** | 0.72 | **0.68** | 0.34 | 0.5 | 0.40 | 0.16 | 0.5 | 0.25 |
| BROAD | 0.60 | 0.66 | **0.52** | **0.61** | **0.67** | 0.52 | 0.41 | 0.5 | 0.45 | 0.10 | 0.5 | 0.16 |
| GEN-DEN | **0.55** | **0.57** | 0.46 | **0.55** | **0.57** | **0.47** | 0.40 | 0.5 | 0.44 | 0.11 | 0.5 | 0.17 |

Table 2: GPT-4 and GPT-3.5 strategy labeling performance over two datasets using expert-annotations shown with majority class classifiers. While performance is moderate, both models significantly outperform majority class classifiers, and GPT-4 outperforms or shows similar performance to GPT-3.5 for all strategies.

| Dataset | Harmful Posts | Counterspeech |
|---|---|---|
| CMV | 1586 | 5191 |
| MTCONAN | 876 | 1000 |

Table 3: Number of unique harmful posts and counterspeech for each dataset.

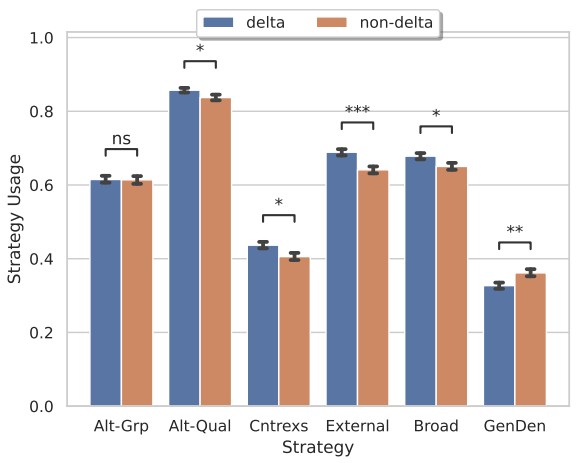

Figure 3: Delta and non-delta comments comparisons of strategy usage annotated with independent $t$-test $P$ values. The legends used in this graph are ns ($p < 1.00$), $\star$ ($p < .05$), $\star\star$ ($p < .01$), and $\star\star\star$ ($p < .001$). Error bars indicate standard error.

(66.7%), BROAD (66.5%), ALT-GRP (61.4%), CNTR-EXS (42.3%), and GEN-DEN (34.3%). On average, each comment used 3.56 strategies.

Using an independent $t$-test, we find that the most significant difference in strategy usage is for the BROAD strategy, with 68.8% and 64.1% for $\Delta$ and non-$\Delta$ comments respectively ($p < 0.001$). We also see that GEN-DEN strategy usage is less persuasive (32.6% in $\Delta$ comments vs 36.1% in non-$\Delta$ comments; $p < 0.01$). Additionally, CNTR-EXS and BROAD were found to be more persuasive.

**Strategies by Demographic Identity Types** The analysis of results categorized by target group demographic types is shown in Figure 5. The keywords corresponding to each demographic group can be found at Appendix H. Among the persuasive

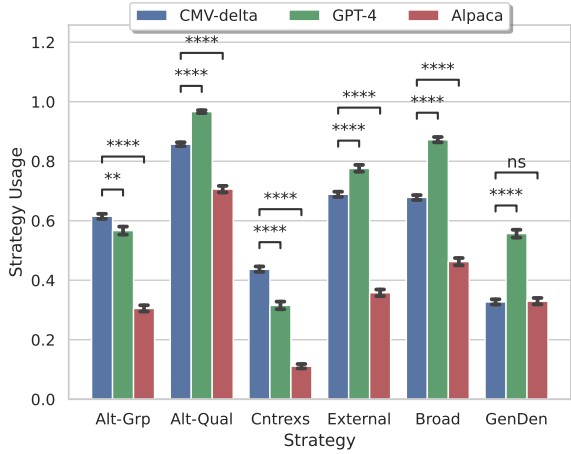

Figure 4: Delta comments and machine-generated counterspeech comparisons of strategy usage annotated with independent $t$-test $P$ values. The legends indicate ns ($p < 1.00$), $\star\star$ ($p < .01$), and $\star\star\star\star$ ($p < .0001$). Error bars indicate standard error.

$\Delta$ comments, the highest proportion of targeted demographic types were related to gender identity, accounting for 42.5% of the total. This was followed by race/ethnicity (10.9%), sexual orientation (8.0%), appearance (7.7%), socio-economic status (4.8%), and religion (4.8%).[7] We notice that the most distinguishing strategies for addressing various demographic types were EXTERNAL and GEN-DEN. Further, GPT-4 showed a higher usage of GEN-DEN when countering stereotypes concerning race/ethnicity, appearance, and religion.

## 6.2 CMV: Persuasive Comments vs. Machine-generated Counterstatements

The comparison between persuasive human responses ($\Delta$ comments) and machine-generated responses in the CMV dataset is shown in Figure 4. Both GPT-4 and Alpaca generated counterspeech showed significant differences in strategy usage compared to persuasive human responses. Alpaca-generated counterspeech contained an average of

---

[7]Some identities are mentioned intersectionally, but in order to holistically evaluate strategy usage associated with identities, we allowed overlaps in our comparison data.

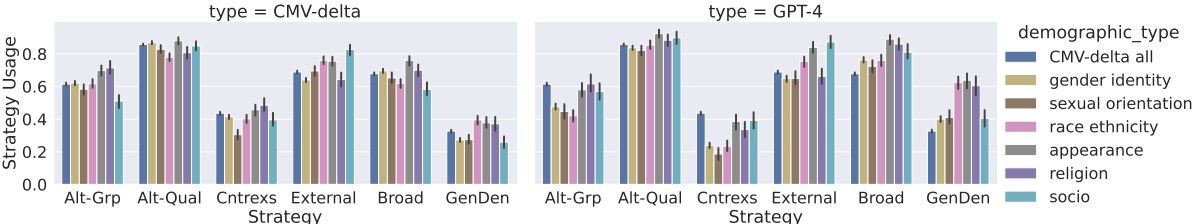

Figure 5: Delta comments and machine-generated counterspeech strategy usage categorized by target group demographic identity types. GPT-4 shows much higher denoucing for certain demographic groups than others.

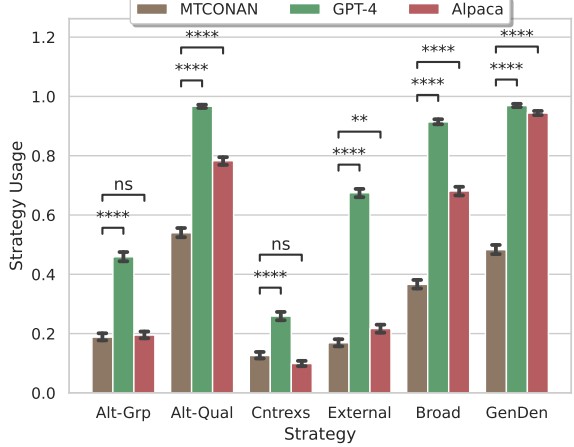

Figure 6: Delta comments and CONAN counterspeech comparisons of strategy usage annotated with independent $t$-test $P$ values. The legends indicate ns ($p < 1.00$), ** ($p < .01$), and **** ($p < .0001$).

2.30 strategies per counterstatement, while GPT-4-generated ones used the most out of the three variations, averaging 4.05 per counterstatement.

Most notably, GPT-4-generated counterspeech contained GEN-DEN more frequently (55.6%) compared to persuasive human (32.6%) and Alpaca-generated (32.3%) responses. Further, both GPT-4 and Alpaca generated counterspeech used CNTR-EXS significantly lesser than human responses (43.6% vs. 31.5% and 11.4% for GPT-4 and Alpaca respectively). Also, GPT-4-generated counterspeech used lower ALT-GRP (56.6%) than human responses (61.4%). The comparison between non-persuasive human responses (non-$\Delta$ comments) and machine-generated responses shown in Appendix C show a similar pattern where GPT-4 uses more GEN-DEN and machines use less CNTR-EXS.

### 6.3 MTCONAN: Counter Narratives vs Machine-generated Counterstatements

The usage of countering strategies in the MTCONAN dataset, compared to machine responses, is presented in Figure 6. In contrast to CMV $\Delta$ comments, the observed trends of counter narra-tives in MTCONAN indicate a higher use of general denoucing and lower usage of all other strategies. The most frequent strategies are ALT-QUAL (54.0%) and GEN-DEN (48.3%), while the least frequent ones are EXTERNAL (16.9%), and CNTR-EXS (12.6%). This analysis reveals that uncivil stereotypical speech is countered differently from civil engagements.

In contrast, machine-generated responses to MT-CONAN posts show high usage of GEN-DEN and ALT-QUAL at 96.9% and 94.4% respectively, significantly differing from MTCONAN counter narratives using 48.3% and 54% respectively. GPT-4-generated responses demonstrate higher usage of all the strategies averaging 4.24 strategies per comment, followed by alpaca (2.91) and MTCONAN counter narratives (1.87).

## 7 Conclusion & Discussion

In this work, we investigated stereotype-targeting strategies that convincingly counter the underlying implication of harmful speech. We crafted six different strategies (§3) and observed human preferences under a controlled setting (§4) and in two datasets (§5). We compared the strategy usage in persuasive vs. unpersuasive human generated counterspeech and in human vs. machine-generated counterspeech (§6).

**Humans prefer more specific strategies.** In both our user study (§4) and data study (§5), ALT-QUAL was the most preferred and used method. Similarly, the delta award difference and the preference ranking both showed EXTERNAL, BROAD, and CNTR-EXS as more convincing. These top strategies, with the exception of broadening, are methods that require specific reasoning about the implication. Moreover, correctly identifying external factors and devise counter examples requires up-to-date world knowledge. In this work, we did not focus on the correctness of the usage of these strategies by machines, but – given their persuasiveness to humans – further study is required to

ensure their quality.

**LLMs use less anti-stereotypical examples.** Across the two models, GPT-4 and Alpaca, we see that LLMs do not use anti-stereotypical examples as often as humans, and instead use more general strategies such as denouncing. However, we see from our results and previous work (Finnegan et al., 2015) that humans prefer countering strategies with specific examples. These findings suggest that LLMs may lack exposure to anti-stereotypical examples and reveals an interesting challenge for improving machine-generated counterspeech.

**Stereotypes about each target group are handled differently.** From both our human evaluations and data analysis, we see different usage of strategies for each stereotype. For example, we observe increased use of EXTERNAL to challenge socio-economic stereotypes (e.g., poor folks, homeless). Similarly, stereotypes about race-ethnicity (e.g., black, person of color, Asian) and appearance (e.g, fat, overweight, slim) also use more EXTERNAL. These results suggest that there are similarities in strategy usage between identity groups that might be associated with stereotypes with similar characteristics. These results show that there is no one correct strategy for counterspeech and a deeper examination is necessary to select strategies with an understanding of the stereotype and group.

**Different intended effects result in different strategy usage.** Our results (§6.1, §6.3) also show that different intended effects (e.g., changing a viewpoint) result in different strategy usage. Although most hate speech datasets are limited to short social media comments (Mathew et al., 2019; Yu et al., 2022), as our Reddit data shows, there are diverse forms of harmful speech that requires countering. Therefore, our findings highlight the importance of further investigation into contextual differences in countering hate speech.

## Limitations & Ethical Considerations

While we attempt a comprehensive understanding of countering strategies through user study and analysis of human and machine-generated counterspeech, we face several limitations as well as unanswered ethical considerations, which are discussed below.

**Human Judgement Variations in Counter-statements** Despite our efforts at creating and evaluating a diverse set of stereotypes and counter-statements with human evaluators, there are still variations in human judgement that we need further investigation. We also only test relatively short counter-statements and limited variations in how the strategies are presented. Therefore, it would be an important step in future works to understand these human variations and account for diversity in both counter-statements and participant pool.

**Human Strategy Evaluation Instructions** To intrinsically motivate and empower annotators, which can increase annotation quality (August et al., 2018), we instructed annotators to select the most convincing statement as a "content moderator". However, this framing could limit the context of convincingness of counter-statements to one group of users whose values and decisions can sometimes disagree with the community members (Weld et al., 2022). Therefore, future works should expand this framing and perform experiments under multiple contexts with varying audiences and intentions, especially focusing on counterspeaker motivations (Buerger, 2022), to understand how these variations interact with strategy effects and preferences.

**Limitations in Strategy Labeling** While we explored different prompts and settings for strategy labeling to improve performance, we did not explore different methods or models. We have also made an effort to create high quality annotations among the authors but were only able to annotate a total of 200 samples across two datasets. This points to a possible further direction into investigating strategy labeling and extraction methods.

**Limitations in Counterspeech Generation** In this work, we only explore two LLMs for counterspeech generation: GPT-4, a closed model which we have limited information about, and Alpaca. Especially with GPT-4, our lack of understanding of training data and procedure leads to unanswered questions about the results shown in this work. We also used naive prompting where we ask the model to "help respond" to the original post to simulate a non-expert interaction with these models. Therefore, future work should explore different settings and prompts including various definitions of counterspeech and countering strategies to improve counterspeech generation.

**Limited Definition of Harm** While counter-speech avoids censoring, it can still have some biases and backfiring effects (Lasser et al., 2023). Additionally, due to differences in perceptions of toxicity (Sap et al., 2022), deciding which posts to respond to could introduce biases, and excessive machine responses could limit conversations. Therefore, future works should strive to understand when these systems should be used to generate counterspeech and how they should be integrated into applications.

**Machine-generated Persuasive Content** Automatic language generation systems have a potential to affect the way users think (Jakesch et al., 2023). Since our main research question asks strategies to persuade readers against harmful beliefs, there is a greater risk of it being used in unintended applications to do the opposite. Additionally, automatically generated counterspeech can be incorrect and therefore can spread false information especially about social groups. Considering these risks, it is important to also investigate into systems and regulations that can protect stakeholders from dangers of malfunctioning or misused systems (Crawford, 2021).

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

## A  Dataset Details

Below sections describe the datasets used in the experiments in detail.

### A.1  Multitarget-CONAN (MTCONAN)

Starting with a seed dataset of human generated hate speech and counterspeech pairs (*N=880*) from NGO workers, Fanton et al. (2021) generate a total of $5,003$ hate speech and counterspeech pairs using human-in-the-loop methods. In particular, they used a machine author and human reviewer paradigm where data was generated iteratively using edits made by NGO workers.

### A.2  Change My View (CMV)

The CMV community on Reddit is "dedicated to civil discourse"[8] and utilizes the delta system to explicitly mark view-changing posts. When a comment changes a user's view, either the original poster or other community members can respond with a "Δ" symbol, marking the comment as *persuasive*. This unique interaction among community members highlights the persuasiveness of various countering strategies and offers valuable insights into their usage and impact on readers.

---

[8]Taken from CMV wiki

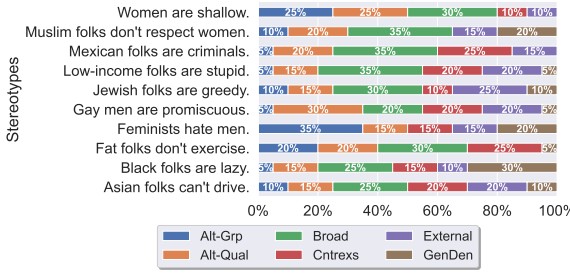

Figure 7: The most convincing strategies by stereotypes.

**(a) Winning Argument Corpus** The Winning Argument Corpus is a dataset of delta and non-delta awarded threads. For each Reddit thread, we consider only the root comment (i.e., direct replies to the original post) to evaluate the arguments used for countering the original post, rather than subsequent interactions. The corpus covers data collected from 2013 to 2015, and to supplement this dataset, we gather additional comments from 2016 to 2022.

**(b) Additional Reddit Data** To select the most relevant subset of the scraped data (2016-2022)[9], we follow these steps:

1. Extract posts (longer than 50 words) with demographic-related words in the title[10], similar to Tan et al.

2. For these posts, retrieve only the (root, Δ-comment) pairs, where a delta has been awarded to the comment[11].

3. Augment this subset with (root, non-Δ comment) pairs, where non-Δ comments are direct replies to the root post that did not receive a delta. For each root post, we select a non-Δ comment based on the highest sentence-BERT similarity (Reimers and Gurevych, 2019) and at least 30% word count overlap with the Δ comment.

Finally, we result in a dataset (referred to as **CMV**) of 5,191 comments across the Winning Argument Corpus (Tan et al., 2016) and the additionally scraped Reddit data.

## B Implied Stereotype Extraction

Our goal is to label posts from both the MTCO-NAN and CMV datasets with the following information: (1) whether the post is harmful towards a demographic group (`harmful_to_group`), (2) if `harmful_to_group`, which demographic group the post is targeting (`target_group`), and (3) if `harmful_to_group`, the stereotypes implied by the post in the form of a generic.

Based on evidence of its labeling abilities (Ziems et al., 2023), we prompt GPT-4 with 3 example posts (two harmful and one non-harmful) to generate these annotations. Following this, we annotate the hate speech posts from the MTCONAN dataset and the titles from the CMV dataset.[12] The results were then validated by three human expert-annotators and the percentage of those marked correct were 99.0%, 98.5%, and 98.5% for `harmful_to_group`, `target_group`, and the generated generics respectively. The detailed settings and prompts are listed in the following sections.

### B.1 GPT-4 Stereotype Extraction Settings

We use the following setting for stereotype extraction mentioned above: `max_tokens=256`, `tempearture=0.7`, and `top_p=1`.

We also use the following system message "`You are evaluating and anlyzing posts found online on whether they contain hate and how they express potentially harmful beliefs. Give all answers in JSON format.`" and the user message shown below filled with the original post.

User message: "`Is the following post harmful towards a demographic group? If so, which groups is the post targeting? Give the implication of the statement in a generic format (e.g., fat folks don't exercise, women are shallow, asian folks can't drive).\n\nPost: "{}"\n\nGive an answer in the following format: {"is_hate_speech": bool, "harmful_to_group": bool, "target_group": str, "implied_generic": str}`"

**Few-shot Examples** We use the following posts as few-shot examples along with the correct labels:

- "CMV: Black people are biologically more prone to violence."

- "Did you hear the one about the jewish philanthropist ? Me neither."

[9]Data source (last accessed 06/23/2023)

[10]The list of words is given in Appendix H

[11]Delta awarded when "!delta" or "Δ" is in the comment, according to the Delta System wiki

[12]We opt for CMV post titles, rather than root posts, as they are more concise and provide a clearer focus for analysis.

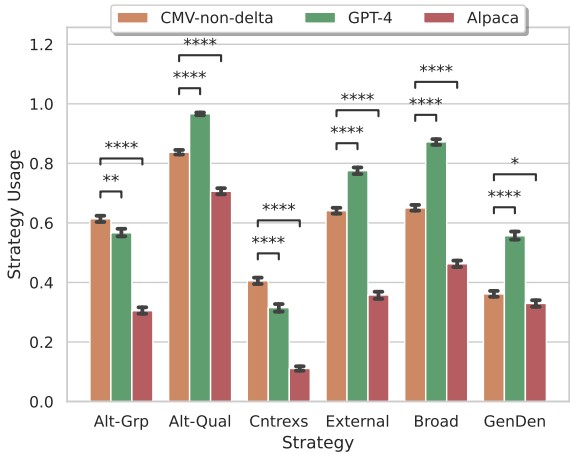

Figure 8: Non-delta comments and machine-generated counterspeech comparisons of strategy usage annotated with independent $t$-test $P$ values. The legends indicate ns ($p < 1.00$), ** ($p < .01$), and **** ($p < .0001$). Error bars indicate standard error.

- "Why? So you can get angry and pretend that all women make false rape claims and having sex with a woman is the biggest risk in the history of mankind?"

The fist two examples are used as harmful posts and the last example is used as a non-harmful post.

## C   Extended Analysis

In addition to our analysis in §6, here we provide a comparison of strategy usage between non-persuasive comments and machine-generated counterspeech and a qualitative analysis of human and machine-generated counterspeech.

### C.1   CMV: Non-persuasive Comments vs Machine-generated Counterstatements

As seen in Figure 8, machine-generated counterspeech shows similar discrepancies mentioned earlier in §6.2. Compared to persuasive (Δ) comments, non-persuasive (non-Δ) comments used more GEN-DEN, but still, GPT-4 used the strategy more significantly than humans whereas Alpaca used it slightly less. CNTR-EXS was used significantly less for both Alpaca and GPT-4.

### C.2   Qualitative Analysis

In Table 4 we show one example of a harmful post from our Reddit dataset. The original post, which claims that "gay men are incapable of a committed relationship", and its implication that, "gay men are promiscuous", is countered by community members with counter examples. However, GPT-4's

repsonse is characterized by longer and frequent denouncing statements and lack of countering examples. Moreover, the broadening method used and labelled by GPT-4 does not actually counter the implication but rather sets the tone (e.g., "let's promote understanding [...] for everyone's individual choices").

Across a number of samples, we observe some common patterns in the usage of countering strategies. First, GEN-DEN has higher occurrence in the beginning or end of the countering posts and is often supplemented with CNTR-EXS or EXTERNAL descriptions as supporting evidence. Additionally, many CNTR-EXS are from first person experiences (e.g., "my gay friends") or the use of second person pronouns (e.g., "you'd get made fun of"). These observations point to future directions in generating counterstatements, including tone setting and directly addressing the reader.

## D   GPT-4 Generation Settings & Prompts

For all generation, we set `max_tokens` to 256 and use `tempearture=0.7`, and `top_p=1`.

### D.1   Counterspeech Generation

We provide the following system message to GPT-4: "You are helping people respond to harmful posts online. Reply directly to the post." along with the following user message where we {post} is replaced with the original post.

User message: "How should I respond to a post that says: "{post}"?"

### D.2   Strategy Labeling

We use the following system message: "You are analyzing countering strategies used in counter arguments of stereotypes expressed as generics. Give all answers in a JSON format." along with user message and few-shot examples that differ by strategy.

The prompts are shown in Table 5. We fill each prompt with counterspeech and implication. Moreover, we use the examples shown in Table 6 and their respective labeling as few shot examples to the model. We use two examples for each strategy labeling task.

## E   Alpaca Generation Setting & Prompts

For counterspeech generation, we used `alpaca-7b` model from huggingface

**Title**: Gay men are incapable of a committed loving, monogamous relationship. CMV

**Implied Stereotype**: Gay men cannot have committed, loving, monogamous relationships

**Post**: This is from my personal anecdotal experiences. There are exceptions, but 98% of gay men prefer a lifestyle of casual, multiple sexual relationships. A partial cause of this is the trauma of being marginalized and demonized by society; but given the choice and freedom to choose, a gay man will overwhelmingly choose a lifestyle of multiple partners over a monogamous committed relationship. I support gay marriage, but our portrayal of a gay family unit is a rarity from what is actually happening, statistically

Δ I'll just chime in on a similar note to others. If you're view is based on anecdotal experience, you should be swayed by the anecdotal experiences of others. I've met lots of gay people, but most of them I honestly don't know what their sexual lifestyle is. But the two close gay friends I have are \*both\* in long-term monogamous relationships. Meanwhile, I know lots of heterosexual friends who at least at this point in their lives seem entirely disinterested in a long term monogamous relationship. So my experiences seem to refute your position.\n\nIn addition, may I ask how you know the gay people you know? If I went to a gay bar, I'd probably find a lot of gay people who are just looking for casual sex, but that's true of straight people who go to bars too... so watch out for confirmation bias.

∇ Of course my evidence is only anecdotal, but I am gay and have been in a committed monogamous relationship for 17 years. That is the longest term relationship in my family, including immediate family who have all been married at least there times. I am also part of several gay themes meetups and am,active in several gay organizations, in all instances, majority of men are partnered and most of those relationships are a decade if not decades long.\n\nAlso, of course you admitted this. It is wildly unfair to judge gay people by criteria that they are not permitted to engage in, namely marriage.

🤖 While I understand that you're sharing your personal anecdotal experiences, it's important not to generalize the experiences and preferences of an entire group based on individual cases. The preferences and choices of individuals within the LGBTQ+ community are as diverse as those in any other community. It's crucial to recognize that people's choices and lifestyles are influenced by various factors, and it's not fair to assume that one's sexual orientation is the primary determinant of their relationship preferences.\n\nMoreover, using a percentage like "98%" without any supporting data or research can be misleading and perpetuate harmful stereotypes. It's essential to remember that people's choices and preferences are personal and should not be used to judge or define an entire community. Instead, let's promote understanding, acceptance, and respect for everyone's individual choices and lifestyles, irrespective of their sexual orientation.

Table 4: Examples of responses for comparison where Δ denotes delta awarded comment, ∇ denotes non-delta comment, and 🤖 denotes GPT-4 generated counterspeech. Highlights were added to example quotes chosen by GPT-4 during each counter strategy labeling task.

---

`chavinlo/alpaca-native` with `max_new_tok` set to 512.

We used the following prompt template where `{input}` is replaced by the post.

Prompt: "`Below is an instruction that describes a task, paired with an input that provides further context. Write a response that appropriately completes the request.`

`### Instruction:`
`You are helping people respond to harmful posts online. Generate a response to the following input.`

`### Input:`
`{input}`

`### Response:`"

## F   Strategy Labeling Error Analysis

An example of a post with its predictions and labels are shown in Table 7. Our research team labelled 1

for strategy ALT-GRP since the comment mentions another group *"an amputee"*; however, GPT-4 did not label this strategy as present.

Moreover, the commenter provides an EXTERNAL reason for misunderstanding of the original poster by pointing out that *"it's typically not so much the male/female roles they're refuting, but that the way their brain sees their body."* This strategy, however, was also predicted to be not present by GPT-4. Such errors highlight the difficulty of the task and model shortcomings.

## G   User Study Details

We use the following settings for user study experiment in §4.

### G.1   Qualification Process & Attention Check

The qualification process included three questions:

1. Humans are mammals.
   How true do you think is the above statement?

2. People are right handed.
   What percentage of people do you think are right handed?

| Strategy | User Message |
|---|---|
| ALT-GRP | Alternate groups are used to challenge generic stereotypes (e.g., women are emotional) and emphasize the presence of various other groups that share similar characteristics (e.g., men can also be emotional). Does the following post include alternate groups to counter the statement that {implied stereotype}?\n\n Post:{counterspeech}\n \n{"alt_groups": bool, "alt_groups_quotes": [str]} |
| ALT-QUAL | Alternate qualities are used to challenge stereotypical generic statements (e.g., feminists hate men) by presenting additional qualities that emphasize more positive or accurate characteristics (e.g., feminists are inclusive, respectful, and supportive of equality for both men and women). Does the following post include alternate qualities to counter the statement that {implied stereotype}?\n\n Post:{counterspeech}\n \n{"alt_qualities": bool, "alt_qualities_quotes": [str]} |
| CNTR-EXS | Counter examples challenge stereotypical generic statements (e.g., women are domestic) by highlighting the existence of numerous instances that contradict such statements (e.g., many women are very career-driven such as women in the medical field, women CEOs, and women professors). Does the following post include counter examples to counter the statement that {implied stereotype}?\n\n Post:{counterspeech}\n \n{"counter_examples": bool, "counter_examples_quotes": [str]} |
| EXTERNAL | External justifications are used to refute stereotypical generic statements (e.g., fat individuals don't engage in exercise) by highlighting external factors that have contributed to the perpetuation of such stereotypes (e.g., media portrayals and beauty standards have reinforced this stereotype, despite the fact that fat individuals do exercise). Does the following post include external reasons to counter the statement that {implied stereotype}?\n\n Post:{counterspeech}\n \n{"external": bool, "external_quotes": [str]} |
| BROAD | Broadening is used to counter stereotypical generic statements (e.g., women are shallow) by questioning the boundaries imposed on the group to demonstrate either the multitude of variations within the group that render the boundary meaningless or the shared characteristics among all humans (e.g., women are complex individuals like everyone else, and some may be shallow but that cannot be generalized to all women). Does the following post use broadening to counter the statement that {implied stereotype}?\n\n Post:{counterspeech}\n \n{"broadening": bool, "broadening_quotes": [str]} |
| GEN-DEN | General denouncing is used to challenge stereotypical generic statements (e.g., feminists hate men) by showing broad disapproval (e.g., this is an unnecessary and hurtful generalization about feminists). Does the following post use general denouncing to counter the statement that {implied stereotype}?\n\n Post:{counterspeech}\n \n{"general_denouncing": bool, "general_denouncing_quotes": [str]} |

Table 5: Prompts used for strategy labeling. We fill {implied stereotype} and {counterspeech} with respective values.

3. Penguins can't fly.
   What percentage of penguins do you think can't fly?

The workers answer using a slider with percentage ranging 0 to 100 or 11 point likert scale. We accept the answer to correct for each question if they answer 1) 10, 2) greater than or equal to $50\%$, and 3) 10. We consider a worker qualified if they score 3 on this task. The workers were paid 0.22 USD for the qualification task. We had 110 participants out of which 95 passed both attention check and choosing different options for most convincing and second most convincing option.

## G.2 Task Instructions

In our task instructions, we asked participants to choose the counter-statement that they found most effective in countering the stereotype.

The exact wording in the instruction was the following: "Pretend you are playing the role of an online content moderator and fact-checker, where your job is to provide counter-statements or corrections when people say things that are stereotypical, generalizations, or blatant biases against certain demographic groups."

Additionally, we provided the following details:

- Determine which of the 6 statements is most effective in counteracting the stereotype by selecting one of 6 options.

- Note: please select the counter-statements that you think are best to counter the stereotype. There is no right answer.

## G.3 Example UI

UI used for human evaluation is shown in Figure 9. We use attention check, which asks the annotator to

| Implication | counterspeech |
|---|---|
| lesbians are butch | "That stereotype is prevalent cause it's the easiest to see. There are tons of lesbians that do not look butch and you wouldn't know without asking their sexuality. To expand this think of the stereotype of gay feminine men. The bears normally aren't outted because they don't look stereotypical most of the time." |
| black folks are violent | "What should be done with this information?\n\nShould police target more black people and put them in jail with less scrutiny? Because that what cops in US already do."
"I mean you are not taking into account all the things that affect testosterone production. Eg smoking is more prevalent among black people than it is with white people.( Smoking increases testosterone fyi) Also talking about blacks as an ethnic group is fucking idiotic since african people are genetically by far more varied than people of european descent." |
| feminists are not willing to acknowledge false accusations and women's capacity for wrongdoing | "No one is saying women never do anything wrong - you're pretty much completely strawmanning the position. They're saying false accusations are rare and unpunished sexual assault is common. None of this has anything to do with accountability, either." |

Table 6: Counterspeech and implications used for few-shot prompting.

| Implied Generic Comment | "Alternative gender identities are invalid and not to be acknowledged"
"How do you feel about transgenderism? From people who I've spoken to about being transgender, it's typically not so much the male/female roles they're refuting, but that the way their brain sees their body and how it ought to feel is different from what it actually is; almost like the way an amputee has phantom limbs, someone who is transgender feels the different parts of their body." | | | | | |
|---|---|---|---|---|---|---|
| Strategies | ALT-GRP | ALT-QUAL | CNTR-EXS | EXTERNAL | BROAD | GEN-DEN |
| Predictions | 0 | 1 | 0 | 0 | 1 | 0 |
| Labels | 1 | 0 | 0 | 1 | 0 | 0 |

Table 7: An example of a comment from CMV with GPT-4 predictions and labels for each strategy.

select the number between 1 to 5 that is randomly populated.

# H  Demographic Identity Related Words

The demographic related words used for data filtering and analysis of countering strategies are shown in Table 8.

## Task

Statement:
**${post}**

This statement implies the following stereotype:
**${stereo}**

It may not be politically correct to say, but how much do you agree that this stereotype is true?

| Strongly disagree | Somewhat disagree | Neither disagree nor agree | Somewhat agree | Strongly agree |

### Counter Statements:

**Counter Statement 1:**
*${example1}*
**Counter Statement 2:**
*${example2}*
**Counter Statement 3:**
*${example3}*
**Counter Statement 4:**
*${example4}*
**Counter Statement 5:**
*${example5}*
**Counter Statement 6:**
*${example6}*

Now choose the **most convincing counter-statement** out of the previous 6 statements:

| **Statement 1**: ${example1} |
| **Statement 2**: ${example2} |
| **Statement 3**: ${example3} |
| **Statement 4**: ${example4} |
| **Statement 5**: ${example5} |
| **Statement 6**: ${example6} |

Please select the fourth option from the left, to double check you're doing this task attentively.

| 1 | 2 | 3 | 4 | 5 |

Now choose the ***second-most*** **convincing counter-statement** out of the remaining statements (excluding the one you selected above):

| **Statement 1**: ${example1} |
| **Statement 2**: ${example2} |
| **Statement 3**: ${example3} |
| **Statement 4**: ${example4} |
| **Statement 5**: ${example5} |
| **Statement 6**: ${example6} |

Figure 9: UI used for human evaluation in §4.

| Demographic Dimension | Terms |
|---|---|
| gender identity | women, men, guys, girls, nonbinary, transgender, cisgender, agender, trans, non-binary, cis, sex, gender |
| intersectional | white men, black women, black men, white women, straight white, queer of color, straight white men, queer white men, queer white women, queer men of color, queer women of color, queer men, queer women, trans of color, cishet white, cisgender heterosexual white, transgender of color, non-disabled white, disabled of color |
| sexual orientation | straight, heterosexual, gay, lesbians, bisexuals, queer, asexual, homosexual, lgbtq, lgbt, monogamous, polyamorous, gay men, lesbian women, butch, bear, femme, feminine |
| age | teenagers, older, millennials, elderly, old, young, middle aged, younger |
| race ethnicity | person of color, people of color, asian americans, asian-americans, asians, asian, african americans, african-americans, black, white-americans, white americans, white, caucasians, hispanic, latinx, latinos, latinas, latin americans, native americans, native, indigenous, native american/first nation, arabs, american indians, alaska native, native hawaiians, pacific islanders |
| nationality | Chinese, Japanese, American, Canadian, Indian, Middle Eastern, European, African, Korean, Mexican, Russian, Cuban, Italian, German, French, Jamaican, Filipino, non-American, foreign, foreigners, Americans |
| religion | buddhists, hindus, christians, jewish, jews, agnostic, muslims, mormons, orthodox, atheists, taoists, protestants, christian, catholics, sikhs, amish, non-religious, satanists, muslim |
| disability | able-bodied, non-disabled, disabled, paralyzed, vision impaired, blind, hearing impaired, deaf, hard of hearing, differently abled, hearing impairment, a visual impairment, vision impairment, disability, paraplegia, quadriplegia, short, cognitive disability, intellectual disability, learning disability, ADHD, a brain injury, autism, autistic, depression, bipolar disorder, psychosocial disability, a mental health condition, mentally disabled, mental illness, mental disorder |
| appearance | tall, short, fat, thin, slim, overweight, bald, ugly, beautiful, light skinned, dark skinned, attractive, unattractive, obese, obesity |
| education | highly educated, less educated, smart, less smart, dumb, stupid |
| personality | introverts, extroverts |
| politics | democrats, libertarians, liberals, republicans, conservatives, feminists |
| socio | rich, wealthy, poor, homeless, lower class, welfare, middle class, working class, upper class, formerly incarcerated, first generation, immigrants, refugees, US citizens |
| country | China, India, the United States, Indonesia, Pakistan, Brazil, Nigeria, Bangladesh, Russia, Mexico, Japan, Ethiopia, Philippines, Egypt, Vietnam, DR Congo, Turkey, Iran, Germany, Thailand, the United Kingdom, France, Italy, South Africa, Myanmar, Kenya, South Korea, Colombia, Spain, Uganda, Argentina, Algeria, Sudan, Ukraine, Iraq, Afghanistan, Poland, Canada, Morocco, Saudi Arabia, Uzbekistan, Peru, Angola, Malaysia, Mozambique, Ghana, Yemen, Nepal, Venezuela, Madagascar, Cameroon, North Korea, Australia, Belgium, Cuba, Greece, Portugal, Sweden, Austria, Israel, Switzerland, Singapore, Norway, Finland, Denmark, New Zealand, Cambodia, Ethiopian, Arabic, Polish, Italians |
| continents | Africa, Asia, the Middle-East, Europe, North America, Central America, South America, Oceania |
| other | racism, sexism |

Table 8: Demographic dimensions and related terms.