# OpenReview forum: "Beyond Denouncing Hate: Strategies for Countering Implied Biases and Stereotypes in Language"
_EMNLP/2023/Conference — EMNLP 2023 Findings_

### Official Review · Reviewer_ds7j · 2023-07-25

**Soundness:** 2

**Excitement:**

3: Ambivalent: It has merits (e.g., it reports state-of-the-art results, the idea is nice), but there are key weaknesses (e.g., it describes incremental work), and it can significantly benefit from another round of revision. However, I won't object to accepting it if my co-reviewers champion it.

**Paper Topic And Main Contributions:**

This paper proposed a list of strategies to challenge the underlying stereotypical implications of hateful language.
They further conducted a user study to examine the convincingness of each strategy.
Finally, they compare the usage in both human and machine-generated CS datasets.

The main contribution is the six strategies that they propose to challenge the stereotypical implications of hate speech.

**Questions For The Authors:**

As mentioned above, I have two questions that may need the authors to answer:
A. Why do you form your tasks as assuming the annotators as content moderators? What standards or rules do content moderators follow in this task?

B. How do you define convincingness and explain it to annotators?

**Reasons To Accept:**

1.This paper is well-written overall.
2.The strategies are sound and well-grounded.
3.Future studies can build on these strategies to develop language technologies to counter hatred.
4.It compares the use of strategies in human and machine generated language.

**Reasons To Reject:**

1. The dataset annotated and validated by humans is small as stated in limitations: only 200.
It is hard for future work to build models upon it. The performance on this dataset is not ideal: for binary classification, f1-scores are between 0.55-0.70. The results for analysis in section 6 are mainly based on the modest classification results, thus lacks convincingness.

2. For the user study based on 200 data, the choice and rationale of settings are not fully justified.
In section 4.1 "annotators are assumed the role of content moderator", while seems intuitive, this may need further justification. What's the rationale to make this assumption for this specific task? What is the standard or guideline that the role of content moderators use or follow? Conflicts are common between moderators and non-moderators (Weld et al., 2022), I believe it is important to consider more individual perspectives. In addition, different communities vary in moderation rules, these have not been considered in this paper too.

Reference: Weld, G., Zhang, A. X., & Althoff, T. (2022, May). What makes online communities ‘better’? Measuring values, consensus, and conflict across thousands of subreddits. In Proceedings of the International AAAI Conference on Web and Social Media (Vol. 16, pp. 1121-1132).

**Reproducibility:**

4: Could mostly reproduce the results, but there may be some variation because of sample variance or minor variations in their interpretation of the protocol or method.

**Reviewer Confidence:**

5: Positive that my evaluation is correct. I read the paper very carefully and I am very familiar with related work.

**Typos Grammar Style And Presentation Improvements:**

Overall this paper is well-structured and reads well.

There are a few typos:

Line 154: multiple "."

Line 362: missing "."

---

> ### Author Rebuttal · Authors · 2023-08-29
>
> Thank you for your thoughtful comments on our work. We appreciate your positive comment about “sound and well-grounded” strategies and your enthusiasm that future studies can build on our work to counter hatred.
>
> 1.  _Human study instructions_: \
> \
> Thank you for providing insightful comments about the human study instructions. We used the wording “role of a content moderator” to motivate our participants [1] to select convincingness in general in a community-agnostic way. As you pointed out, there can be disagreements between moderators and community [2]. However, asking annotators to assume the role of content moderator, someone who is often a volunteer community leader, over the role of a community member encourages the annotators to think directly about what they might write to the poster while also keeping community well-being in mind. There may be certain strategies that are more effective in reducing disagreements between community members or that are successful in certain communities (e.g., subreddits have their own social and communication norms [2]). We will add a discussion of this in the paper to be explored by future work. See response to Reviewer L28W for further details.
> \
> \
> [1] Tal August, Nigini Oliveira, Chenhao Tan, Noah Smith, and Katharina Reinecke. 2018. [Framing Effects: Choice of Slogans Used to Advertise Online Experiments Can Boost Recruitment and Lead to Sample Biases](https://doi.org/10.1145/3274291). Proc. ACM Hum.-Comput. Interact. 2, CSCW, Article 22 (November 2018), 19 pages.\
> [2] Weld, G., Zhang, A. X., & Althoff, T. (2022, May). [What makes online communities ‘better’? Measuring values, consensus, and conflict across thousands of subreddits](https://doi.org/10.48550/arXiv.2111.05835). In Proceedings of the International AAAI Conference on Web and Social Media (Vol. 16, pp. 1121-1132).
> \
> &nbsp;
> 2.  _Classification performance_: \
> \
> We do agree that the results seem low at first glance. To clarify our strategy labeling performance by GPT-4, we would like to highlight that our performance is comparable to previous works in various argumentative strategies detection: persuasive strategy detection in meme text (0.593 micro F1) [1],  persuasive strategies in English news (0.376 micro F1) [2], propaganda metaphor detection (single label) in news (0.387 F1 for loaded language and 0.287 F1 for name calling) [3], logical fallacy detection (0.587 F1 score for fine-tuned and 0.136 for GPT-2 zero shot) [4]. Collectively these results show that argumentative strategy labeling is a hard task, even for fine-tuned models, and that our performance is within range of these models despite not fine-tuning (instead using few-shot prompting). \
> \
> Since argument strategy labeling is a hard task, as also shown by previous works mentioned above, we needed expert annotations. Since we wanted to measure the quality of our GPT-4-based classifier, we focused on obtaining a high-quality and high-agreement test set (by resolving disagreements through deliberation between expert annotators for all samples, which were often long posts) rather than trying to obtain more annotations. Test sets of randomly sampled 100-200 examples are not uncommon for validation [5], and since our objective is to validate the usability of the current model rather than to train a new model, it is unlikely that increasing our test set size would impact our results. \
> \
> Nevertheless, we do acknowledge that there is a room for improvement in strategy labeling, which will be an interesting future work (see Limitations in Strategy Labeling under Section Limitations & Ethical Considerations). To provide insights into how future work can improve upon this classification task, we will add other baseline models' performance: ChatGPT (`gpt-3.5-turbo-0314`) and majority class classifiers. As shown in the table below, GPT-4 outperforms other methods in strategy labeling in all but two strategies, ALT-QUAL and BROAD. Additionally, we will include error analysis of GPT-4’s classification performance in the final version of our paper. \
> &nbsp;
> |      Method    |          GPT-4		|||ChatGPT           |||          Majority Classifier (0)		|||          Majority Classifier (1)		|||
> | -------- | -------- | -------- | -------- | -------- | -------- | -------- | -------- | -------- | -------- | -------- | -------- | -------- |
> | Strategy | Precision | Recall | F1 | Precision | Recall   | F1   |Precision | Recall   | F1   |Precision | Recall   | F1   |
> | ALT-GRP  | **0.59**  | **0.65** | **0.56** | 0.56     | 0.61   | 0.55     | 0.42 | 0.5 | 0.45 | 0.09 | 0.5 | 0.15 |
> | ALT-QUAL | **0.63**  | **0.61** | 0.55     | 0.60     | 0.61   | **0.61** | 0.31 | 0.5 | 0.38 |0.20| 0.5 | 0.28 |
> | CNTR-EXS | **0.70**  | **0.72** | **0.70** | 0.54     | 0.54   | 0.54     | 0.36 | 0.5 | 0.42 | 0.14 | 0.5 | 0.22 |
> | EXTERNAL | **0.69**  | **0.71** | **0.69** | 0.62     | 0.63   | 0.63     | 0.34 | 0.5 | 0.40 | 0.16 | 0.5 | 0.25 |
> | BROAD    | 0.63      | **0.70** | 0.57     | **0.64** | 0.68   | **0.65** | 0.41 | 0.5 | 0.45 | 0.10 | 0.5 | 0.16 |
> | GEN-DEN  | **0.61**  | **0.66** | **0.59** | 0.56     | 0.59   | 0.56     | 0.40 | 0.5 | 0.44 | 0.11 | 0.5 | 0.17 |
>
>       [1] Dimitar Dimitrov, Bishr Bin Ali, Shaden Shaar, Firoj Alam, Fabrizio Silvestri, Hamed Firooz, Preslav Nakov, and Giovanni Da San Martino. 2021. [SemEval-2021 Task 6: Detection of Persuasion Techniques in Texts and Images. In Proceedings of the 15th International Workshop on Semantic Evaluation (SemEval-2021)](https://aclanthology.org/2021.semeval-1.7), pages 70–98, Online. Association for Computational Linguistics.\
>       [2] Jakub Piskorski, Nicolas Stefanovitch, Giovanni Da San Martino, and Preslav Nakov. 2023. [SemEval-2023 Task 3: Detecting the Category, the Framing, and the Persuasion Techniques in Online News in a Multi-lingual Setup](https://aclanthology.org/2023.semeval-1.317). In Proceedings of the 17th International Workshop on Semantic Evaluation (SemEval-2023), pages 2343–2361, Toronto, Canada. Association for Computational Linguistics.\
>       [3] Daniel Baleato Rodríguez, Verna Dankers, Preslav Nakov, and Ekaterina Shutova. 2023. [Paper Bullets: Modeling Propaganda with the Help of Metaphor](https://aclanthology.org/2023.findings-eacl.35). In Findings of the Association for Computational Linguistics: EACL 2023, pages 472–489, Dubrovnik, Croatia. Association for Computational Linguistics.\
>       [4] Zhijing Jin, Abhinav Lalwani, Tejas Vaidhya, Xiaoyu Shen, Yiwen Ding, Zhiheng Lyu, Mrinmaya Sachan, Rada Mihalcea, and Bernhard Schoelkopf. 2022. [Logical Fallacy Detection](https://doi.org/10.48550/arXiv.2202.13758). In Findings of the Association for Computational Linguistics: EMNLP 2022, pages 7180–7198, Abu Dhabi, United Arab Emirates. Association for Computational Linguistics.\
>       [5] Nikita Nangia, Clara Vania, Rasika Bhalerao, and Samuel R. Bowman. 2020. [CrowS-Pairs: A Challenge Dataset for Measuring Social Biases in Masked Language Models](https://aclanthology.org/2020.emnlp-main.154). In Proceedings of the 2020 Conference on Empirical Methods in Natural Language Processing (EMNLP), pages 1953–1967, Online. Association for Computational Linguistics.\
>       &nbsp;
> 3. _Definition of convincingness_: \
> \
> Thank you for asking a question about an important point of clarification: the definition of convincingness provided to human study participants. In our detailed instructions, we asked participants to choose the counter-statement that they found most effective in countering the stereotype.\
> \
> The exact wording in the instruction was the following:
>
>    - Determine which of the 6 statements is most effective in counteracting the stereotype by selecting one of 6 options.
>    - Note: please select the counter-statements that you think are best to counter the stereotype. There is no right answer.
>
> &nbsp;&nbsp;&nbsp;&nbsp;&nbsp;&nbsp;&nbsp;&nbsp;We will add more details about the specific instructions in the Appendix of the final version.

---

### Official Review · Reviewer_PBEi · 2023-08-03

**Soundness:** 3

**Excitement:**

4: Strong: This paper deepens the understanding of some phenomenon or lowers the barriers to an existing research direction.

**Paper Topic And Main Contributions:**

This paper focuses on the stereotypical implications of hateful language. They found that naturally written counterarguments use countering strategies more specific to the implied stereotype than machine-generated counterarguments; machine-generated counterspeech uses less specific strategies. They compared the countering strategies used by humans versus the ones generated by machines (GPT-4 and Alpaca). This was done to understand whether machines use countering strategies as humans. They found that less complex countering strategies are not necessarily the most convincing counterarguments.


**Questions For The Authors:**

Q1- Why these six strategies? (line 237: "crafted based on prior work on combating stereotypes" ). Which prior work do these strategies come from?


**Reasons To Accept:**

- Analyze an interesting problem related to counterarguments
- Extensive analysis
- Well-written paper and easy to follow

**Reasons To Reject:**

- No substantial reason to reject.

**Reproducibility:**

4: Could mostly reproduce the results, but there may be some variation because of sample variance or minor variations in their interpretation of the protocol or method.

**Reviewer Confidence:**

4: Quite sure. I tried to check the important points carefully. It's unlikely, though conceivable, that I missed something that should affect my ratings.

**Typos Grammar Style And Presentation Improvements:**

- Line 362: missing period.

---

> ### Author Rebuttal · Authors · 2023-08-29
>
> Thank you for your positive review. We are excited that you found the topic of our work an “interesting problem related to counter argument”.
>
> 1.  _Strategy selection_: \
> \
> Our six strategies were selected based on preliminary investigations on existing hate speech datasets, as well as by drawing from related work in NLP [1] and psychology and philosophy. As discussed in Section 2.1, Language and Stereotype, generics, i.e., generalizing statements such as “girls wear pink” [2], often convey stereotypes about social groups and lead to default assumptions and generalization [3]. With this in mind, we constructed the strategies that challenge specific qualities of generics such as typicality (generalizability), definitional quality [4], and striking quality [5] (see Section 3 for further details). Additionally, we included general denouncing, which is more commonly seen in counterspeech literature [6]. We acknowledge that there may be other strategies that could be useful in countering social stereotypes, which we hope future work will explore.
> \
> \
> [1] Emily Allaway, Nina Taneja, Sarah-Jane Leslie, and Maarten Sap. 2023. [Towards countering essentialism through social bias reasoning](https://doi.org/10.48550/arXiv.2303.16173).\
> [2] Marjorie Rhodes, Sarah-Jane Leslie, and Christina M. Tworek. 2012. [Cultural transmission of social essentialism](https://doi.org/10.1073/pnas.1208951109). Proceedings of the National Academy of Sciences, 109(34):13526–13531.\
> [3] Sarah-Jane Leslie. 2014. [Carving up the social world with generics](https://philpapers.org/rec/LESCUT). Oxford studies in experimental philosophy, 1.\
> [4] Robert Rooij and Katrin Schulz. 2020. [Generics and typicality: A bounded rationality approach](https://doi.org/10.1007/s10988-019-09265-8). 43(1):83–117.\
> [5] Sarah-Jane Leslie. 2017. [The Original Sin of Cognition: Fear, Prejudice, and Generalization](https://doi.org/10.5840/jphil2017114828). 114(8):393–421.\
> [6] Maxime Charles Lepoutre. 2019. [Can ’More Speech’ Counter Ignorant Speech? Tackling the Stickiness of Verbal Ignorance](https://doi.org/10.26556/jesp.v16i3.682). 16(3).

---

### Official Review · Reviewer_L28W · 2023-08-07

**Typos Grammar Style And Presentation Improvements:** Line 87 - humna-written
**Soundness:** 4

**Excitement:**

4: Strong: This paper deepens the understanding of some phenomenon or lowers the barriers to an existing research direction.

**Paper Topic And Main Contributions:**

The paper proposes countering hateful content by considering its implied stereotypes and biases towards the target group. 6 strategies, i.e., alternate groups, alternate qualities, counterexamples, external factors, broadening, and general denouncing, are composed to do so, and human and machine generated counter speech datasets are analysed in terms of strategy usages. The results show that while automatically generated counter speeches are more generic and denouncing, mostly towards the hatred itself, the human-written ones are mostly targeting the implied statements and more specific. In addition, machine generated counter speech is found less convincing in comparison to the human-written replies.

**Questions For The Authors:**

How many samples are used in Section 4? It sounds like a cumbersome task to craft six counter-statements for many samples by all the authors.

Line 329: How did you select 10 groups among the top 25 and what were these groups? This section needs more clarifications.

"the role of a content moderator" does not imply that the selected counter speech would be the most convincing. I disagree that one can accurately evaluate the effectiveness of a counterspeech from the provider point of view. Why didn't you use the role 'bystander', who is the most critical group to convince and represents the receiver of counterspeech?

It would be fairer if the LLMs were given actual instructions to respond to harmful posts. Suppressing the real power of an LLM and comparing it with human generated text do not yield a proper conclusion. The authors did not even employ implied biases in the generation process. It is not clear what was tried to be proven by comparing:
Human-written, their convincingness already proven replies with machine-written no specific instruction-provided replies. There are certain behaviours of these models, regarding biases inherently coming from their training data. If there was a comparison among non-persuasive CMV comments and machine generated counterspeech, then we most probably would see a different picture. If it is a wrong assumption, I would like to read the reasoning of the authors.

**Reasons To Accept:**

The experimentations on convincingness and effectiveness of the counter speech is very valuable. The paper is well-written and easy to follow. Extensive discussion and in-depth analyses of the results are provided.

**Reasons To Reject:**

The role of stereotype or implied bias in the final results, especially when the generation task results are discussed, is not clear. A few very important numbers are missing: e.g., the number of samples that the authors wrote a counter-speech for each 6 strategies.

**Reproducibility:**

2: Would be hard pressed to reproduce the results. The contribution depends on data that are simply not available outside the author's institution or consortium; not enough details are provided.

**Reviewer Confidence:**

4: Quite sure. I tried to check the important points carefully. It's unlikely, though conceivable, that I missed something that should affect my ratings.

---

> ### Author Rebuttal · Authors · 2023-08-29
>
> Thank you for your thoughtful response and for considering our "experimentations on convincingness and effectiveness of the counter speech [as] very valuable." Here is our response to your questions.
>
> 1.  _Counterspeech generation process_: \
> \
> We will clarify how we instructed the LLMs: we used the system message / instruction “You are helping people respond to harmful posts online. Reply directly to the post.” and user message / input “How should I respond to a post that says: [POST]?” (Appendix D1). By instructing it as such, we wanted to avoid biasing the models' responses (e.g., prompting models to focus on implied biases which could lead them to not respond to the post's content, or prompting models to only use one specific strategy).
> \
> \
> We made this instruction somewhat general not to suppress LLMs but rather to generate a sample that is unbiased by instructions. The goal of this paper was not to study prompting approaches, which can be an interesting future work (see Limitations in Counterspeech Generation under Limitation & Ethical Consideration). Instead, our goal was to understand how the models approach responding to harmful posts, especially in strategy usage patterns in their responses.
> \
> \
> We will move more specific prompting instructions into the main part of the paper in our final version.
> \
> &nbsp;
> 2. _Comparing machine-generated responses to human-written ones_: \
> \
> We chose to compare the machine generated output with human generated content with “proven convincingness” in order to directly address the main goal of the paper: investigating effective strategies for countering implied bias and stereotypes. To do this, we first compared convincing and unconvincing human-written responses (delta to non-delta CMV reddit posts), in order to identify patterns in how humans used different strategies to convince. Then, to understand the strategy choices of LLMs, along with the convincingness of LLM generated counterspeech, we compared directly to only the convincing human-written counterspeech. In this way, we wanted to gain an understanding of how effective LLMs might be as tools for counterspeaking or when interacting with users with stereotypical beliefs with suggested directions of improvements.
> \
> \
> While comparing machine generated results to non-persuasive comments may not directly address the main goal of our paper, we agree that it would yield an interesting comparison and will add it to the Appendix section in the final version.
> \
> &nbsp;
> 3.  _Human study instructions_: \
> \
> It is a great point that the framing of the human convincingness evaluation task is an important factor. In our task instructions, we used the wording “the role of a content moderator” so that participants would be more intrinsically motivated and empowered to action (which increases annotation quality [6]) and by doing so, focus on choosing convincing answers that the participants might have written themselves as an agentic and active community member.
> \
> \
> While we do agree that counterspeakers often consider bystanders as the most critical group to convince [1], we did not want to specifically direct the question of convincingness to bystanders when the goal of counterspeech and the meaning of effective counterspeech are still open questions [2]. Bystanders are also less likely to engage [3], and for our study, we wanted the participants to have greater agency when considering both community well-being and the poster. This gives us a better understanding of the intent of the counterspeaker, the participants in this case, when making their choices.
> \
> \
> Moreover, content-moderators, especially in communities such as Reddit and Wikipedia, are volunteer community members that take on multiple community roles and use many community moderation approaches beyond just flagging and removing posts, much like community leaders [4]. Therefore, we think that by asking the participants to assume the role of a content moderator, our results can both help with explanations of moderation decisions [5] and engagement as a community member, i.e., bystander.
> \
> \
> Nevertheless, we agree that the question of audience selection and strategy variation is an interesting future work, which we will expand on in our discussion and limitations section in the final version.
> \
> \
> [1] Catherine Buerger. 2022. [Why They Do It: Counterspeech Theories of Change](https://dx.doi.org/10.2139/ssrn.4245211).\
> [2] Susan Benesch, Derek Ruths, Haji Mohammad Saleem, Kelly P. Dillon, and Lucas Wright. 2016. [Considerations for Successful Counterspeech](https://dangerousspeech.org/wp-content/uploads/2016/10/Considerations-for-Successful-Counterspeech.pdf).\
> [3] Samuel Hardman Taylor, Dominic DiFranzo, Yoon Hyung Choi, Shruti Sannon, and Natalya N. Bazarova. 2019. [Accountability and Empathy by Design: * Encouraging Bystander Intervention to Cyberbullying on Social Media](https://doi.org/10.1145/3359220). Proc. ACM Hum.-Comput. Interact. 3, CSCW, Article 118 (November 2019), 26 pages.\
> [4] Joseph Seering. 2020. [Reconsidering Community Self-Moderation: the Role of Research in Supporting CommunityBased Models for Online Content Moderation](https://doi.org/10.1145/3415178). Proc. ACM Hum.-Comput. Interact. 4, CSCW2, Article 107 (October 2020), 28 pages.\
> [5] Shagun Jhaver, Amy Bruckman, and Eric Gilbert. 2019. [Does Transparency in Moderation Really Matter?: User Behavior After Content Removal Explanations on Reddit](https://doi.org/10.1145/3359252). Proc. ACM Hum.-Comput. Interact. 3, CSCW, Article 150 (November 2019), 27 pages.\
> [6] Tal August, Nigini Oliveira, Chenhao Tan, Noah Smith, and Katharina Reinecke. 2018. [Framing Effects: Choice of Slogans Used to Advertise Online Experiments Can Boost Recruitment and Lead to Sample Biases](https://doi.org/10.1145/3274291). Proc. ACM Hum.-Comput. Interact. 2, CSCW, Article 22 (November 2018), 19 pages.\
> &nbsp;
> 4. _Missing numbers_: \
> \
> Thank you for pointing out the missing numbers! We will include them in the final version of our paper.
> \
> \
> In section 4, our research team crafted six counter-statements for each of the 10 toxic posts that targeted different groups, a total of 60 counter-statements. This process took our research team multiple revisions and pilot experiments with different wordings.
> \
> \
> The top 25 groups were ranked by frequency, i.e., the number of mentions in the field “target group” in the original dataset (SBIC; [1]), and were aggregated by a rough regex matching identity terms such as “asian [folks|people|person]” or “asians” within the target group identified by the SBIC annotators.
> \
> \
> [1] Maarten Sap, Saadia Gabriel, Lianhui Qin, Dan Jurafsky, Noah A Smith, and Yejin Choi. 2020. [Social bias frames: Reasoning about social and power implications of language](https://aclanthology.org/2020.acl-main.486). In ACL.

---

### Meta-Review · Area_Chair_V6gq · 2023-09-14

**Recommendation:** 3

**Metareview:**

Reviewers thought that this paper was interesting and covered an important topic. They thought that the taxonomy of proposed strategies was well-motivated, and could be useful to build on in future work; some also found the annotation and LLM-enabled analyses valuable as well. Taken together, my sense is that the paper makes an interesting contribution to the area of NLP-enabled counterspeech, but could be substantially improved. I’ve highlighted some concerns from the reviews, and I encourage the authors to attend to these, as well as to the rest of the reviewers’ comments, as they revise their paper.
* The notion of “convincingness” is key to the paper, and reviewers had doubts about whether the authors operationalized it in a sound way. I’d further add that “convincingness” means entirely different things in the context of CMV and in the instructions to annotators. This merits clarification: it is find for the authors to examine multiple notions of convincingness, but they should be more explicit that their operationalization is narrow, and perhaps tailored to specific purposes that they could elaborate on (e.g., certain forms of moderation).
* The classifier performance is low. We realize this is because the task is hard, but the low performance, barring any further explanation, seems to undermine much of the analyses in section 6. For instance, if classification errors, as the authors say, often come from “mere mentions of alternative qualities/groups” that do not have a countering nature, then this seems like a systematic overestimation. Here, the authors could either improve their classifier — perhaps by increasing their data size, as reviewers suggest, or more critically discuss the results of their analysis.

---

### Decision · Program_Chairs · 2023-10-07

**Decision:**

Accept-Findings

**Comment:**

Reviewers thought that this paper was interesting and covered an important topic. They thought that the taxonomy of proposed strategies was well-motivated, and could be useful to build on in future work; some also found the annotation and LLM-enabled analyses valuable as well. Taken together, my sense is that the paper makes an interesting contribution to the area of NLP-enabled counterspeech, but could be substantially improved. I’ve highlighted some concerns from the reviews, and I encourage the authors to attend to these, as well as to the rest of the reviewers’ comments, as they revise their paper.
* The notion of “convincingness” is key to the paper, and reviewers had doubts about whether the authors operationalized it in a sound way. I’d further add that “convincingness” means entirely different things in the context of CMV and in the instructions to annotators. This merits clarification: it is find for the authors to examine multiple notions of convincingness, but they should be more explicit that their operationalization is narrow, and perhaps tailored to specific purposes that they could elaborate on (e.g., certain forms of moderation).
* The classifier performance is low. We realize this is because the task is hard, but the low performance, barring any further explanation, seems to undermine much of the analyses in section 6. For instance, if classification errors, as the authors say, often come from “mere mentions of alternative qualities/groups” that do not have a countering nature, then this seems like a systematic overestimation. Here, the authors could either improve their classifier — perhaps by increasing their data size, as reviewers suggest, or more critically discuss the results of their analysis.